# Generation of a Transgenic *Plasmodium cynomolgi* Parasite Expressing *Plasmodium vivax* Circumsporozoite Protein for Testing *P. vivax* CSP-Based Malaria Vaccines in Non-Human Primates

**DOI:** 10.3390/vaccines13050536

**Published:** 2025-05-17

**Authors:** Maya Aleshnick, Shreeya Hegde, Charlie Jennison, Sebastian A. Mikolajczak, Ashley M. Vaughan, Derek Haumpy, Thomas Martinson, Judith Straimer, Brandon K. Wilder

**Affiliations:** 1Vaccine and Gene Therapy Institute, Oregon Health and Science University, Beaverton, OR 97006, USA; aleshnic@ohsu.edu (M.A.); haumpy@ohsu.edu (D.H.); martitho@ohsu.edu (T.M.); 2Global Health, Biomedical Research, Novartis, Emeryville, CA 94608, USA; shreeya.hegde@novartis.com (S.H.); samikola@yahoo.ca (S.A.M.); judith.straimer@novartis.com (J.S.); 3Center for Global Infectious Disease Research, Seattle Children’s Research Institute, Seattle, WA 98101, USA; charles.jennison@biontech.de (C.J.); ashley.vaughan@seattlechildrens.org (A.M.V.)

**Keywords:** *Plasmodium vivax*, *Plasmodium cynomolgi*, pre-erythrocytic malaria vaccines, circumsporozoite protein, hypnozoite

## Abstract

**Background/Objectives**: Malaria, caused by infection with *Plasmodium* parasites, exacts a heavy toll worldwide. There are two licensed vaccines for malaria as well as two monoclonal antibodies that have shown promising efficacy in field trials. The vaccines and monoclonal antibodies target the major surface protein (circumsporozoite protein, CSP) of *Plasmodium falciparum*. Yet *P. falciparum* is only one of the four major species of *Plasmodium* that infect humans. *Plasmodium vivax* is the second leading cause of malaria, but the *P. vivax* vaccine and monoclonal development lags far behind that for *P. falciparum* owing to the lack of basic preclinical tools such as in vitro culture or mouse models that replicate the key biological features of *P. vivax*. Notably among these features is the ability to form dormant liver stages (hypnozoites) that reactivate and drive the majority of the *P. vivax* malaria burden. *Plasmodium cynomolgi* is a simian parasite which is genotypically very close and phenotypically similar to *P. vivax*; it can infect non-human primates commonly used in research and replicates many features of *P. vivax*, including relapsing hypnozoites. **Methods**: Recently, a strain of *P. cynomolgi* has been adapted to in vitro cultures allowing parasite transgenesis. Here, we created a transgenic *P. cynomolgi* parasite in which the endogenous *P. cynomolgi* CSP has been replaced with *P. vivax* CSP, with the goal of enabling the preclinical study of anti-*P. vivax* CSP interventions to protect against primary and relapse infections. **Results**: We show that the in vitro-generated transgenic *Pcy*[*Pv*CSP] parasite expresses both serotypes of *P. vivax* CSP and retains full functionality in vivo, including the ability to transmit to laboratory-reared *Anopheles* mosquitoes and cause relapsing infections in rhesus macaques. To our knowledge, this is the first gene replacement in a relapsing *Plasmodium* species. **Conclusions**: This work can directly enable the in vivo development of anti-*P. vivax* CSP interventions and provide a blueprint for the study of relapsing malaria through reverse genetics.

## 1. Introduction

Malaria remains one of the most impactful infectious diseases worldwide, with approximately 263 million cases across 83 malaria-endemic countries in 2023 causing an estimated 597,000 deaths [1]. While the total number of malaria deaths fell precipitously from 2000 to 2015, this progress has plateaued in recent years [2], highlighting the urgency of developing new malaria interventions, including vaccines and therapeutics. Parasites in the genus *Plasmodium* are the causative agent of malaria, and infection of the vertebrate host is initiated when a female *Anopheles* mosquito injects the sporozoite stage of the parasite into the skin while probing for a blood meal. These parasites travel to the liver, where each parasite infects a single hepatocyte and over ~7–10 days forms tens of thousands of merozoites. These merozoites are released into the blood where iterative rounds of replication in red blood cells result in disease symptoms and transmission to the next vector (lifecycle reviewed by Simwela and Waters [3]). Several species of *Plasmodium* are responsible for the morbidity and mortality attributed to malaria, notably *P. falciparum (Pf)* and *P. vivax (Pv)*. While *Pf* is responsible for the majority of malaria-associated deaths worldwide [2], *Pv* is the most widespread and exerts a significant toll in endemic areas, with nearly five million cases estimated in 2021 [2]. In addition, *Pv* poses a unique challenge for elimination efforts due to the dormant hypnozoites that form in the liver which can reactivate months to years after the initial infectious mosquito bite [4]. These hypnozoites are a primary driver of disease and transmission as they are the source of ~80% or more of *Pv* blood-stage infections [5,6].

While several points of the parasite lifecycle provide potential vaccine targets [7], intervening at the pre-erythrocytic stage is an especially promising strategy as this stage is a bottleneck with relatively small numbers of extracellularly exposed parasites [8]. Abrogating parasite development at this stage would prevent all disease and the transmission associated with the succeeding blood stage. Indeed, the only licensed malaria vaccines to date, RTS, S/AS01 (Mosquirix^TM^), and R21/Matrix-M^TM^, both contain a pre-erythrocytic stage antigen [9]. These vaccines both rely on antibodies against the circumsporozoite protein (CSP)—an abundant, essential protein with numerous critical functions during development in the mosquito and infection of the liver [10,11,12,13,14,15,16]. The general structure of CSP is conserved across *Plasmodium* species [15]; however, the immunodominant “central repeat” region that is the target of neutralizing antibodies is variable between *Plasmodium* species with no serological overlap [17]. Thus, any CSP-based intervention must be developed specifically for each parasite species. In contrast to the two CSP-based *Pf* vaccines and the promising monoclonal antibodies (mAbs), CIS43LS and L9LS, that are in clinical development [18,19,20], there is a pronounced lag in the development of similar CSP-based interventions for *Pv* [21].

The development of interventions against *Pv* has been hindered by the technical challenges of studying this parasite, namely the inability to culture the erythrocytic stages in vitro as is routinely performed for *Pf* [22]. The result is that, aside from a few notable experimental infections of humans [23], the study of *Pv* parasites at any stage must start by isolating blood from naturally infected individuals. In addition, unlike the *Pf* CSP repeat region, which is nearly invariant across field isolates, *Pv* has at least two major and one minor CSP repeat serotype [24], and so any CSP-based *Pv* vaccine will need to be at least bivalent to provide protection against the majority of circulating parasites. These difficulties in developing *Pv* interventions are exacerbated by the lack of an in vivo model to study *Pv* liver-stage biology, a better understanding of which will be critical for controlling and eradicating the disease.

While the murine *Plasmodium* species have allowed for substantial advances in *Plasmodium* research and the development of interventions, there is no mouse model of hypnozoites in which to study this critical parasite reservoir [22,25,26,27]. The logistical challenges in testing interventions in a controlled human malaria infection, as is commonplace for *Pf* [28], leave gaps at every step of the development pipeline [29]. However, the simian malaria parasite *P. cynomolgi* is phylogenetically very closely related to *Pv* [30] and mimics key aspects of *Pv* biology, including a preference for reticulocytes at the blood stage [3], the ability to form relapsing hypnozoites [3,31], and the infection of humans [32,33]. Importantly, *Pcy* infects common laboratory non-human primates (NHPs), including rhesus macaques (RMs), and the entire *Pcy* life cycle can be completed between NHPs and laboratory strains of *Anopheles* mosquitoes—allowing consistent studies of infection and relapse. Thus, *Pcy* has been a cornerstone of *Pv* research [3,25,31], and, indeed, hypnozoites were first described for *Pcy* [34] before their identification in *Pv* (reviewed by Voorberg-van der Wel et al. [27]). This model was also critical in the identification of primaquine as the first pharmaceutical against liver-stage hypnozoites [35] and has been used in the study of several vaccines against *Pv* [36,37,38,39,40,41,42,43,44]. The first vaccination study using a *Pcy* sporozoite challenge showed this model to be stringent, as a viral-vectored vaccine showed no efficacy [34] and a live-attenuated vaccination provided a low but partial immunity [45], reflecting what was observed with similar vaccines against *Pf* in humans. While the *Pcy* animal model is ideal in many ways, the immunodominant repeat regions of *Pcy* CSP and *Pv* CSP are divergent [46], which precludes the direct assessment of *Pv* CSP-based interventions with an antibody component using *Pcy* in NHPs. Recently, *Pcy* asexual blood stages have been culture-adapted [47], which now allows for genetic manipulation without the need for propagation in NHPs, as demonstrated by a recent study that introduced a drug resistance marker [48]. However, these in vitro parasites have not yet been directly transmitted to mosquitoes as is commonplace for *Pf*.

In this study, we aimed to overcome these practical limitations by creating a *Pcy* parasite which expresses the *Pv* CSP and can complete the parasite life cycle between laboratory-reared *Anopheles* mosquitoes and NHPs. This transgenic parasite fills a critical gap in the development pipeline of *Pv* pre-erythrocytic stage interventions. Furthermore, our study provides the first example, to our knowledge, of gene replacement in *Pcy* parasites which provides a blueprint for studies of relapsing *Plasmodium* parasites in vivo through reverse genetics.

## 2. Materials and Methods

### 2.1. Optimization of In Vitro Parasite Culturing Conditions

*Macaca mulatta* blood and serum was supplied by BioIVT (Westbury, NY, USA) from US-born animals that had not been treated with antibiotics for at least one year. Samples were collected in a sterile manner with needles switched between extraction and transfer and vacutainer. Whole blood was collected in heparin tubes (BD Biosciences, Franklin Lakes, NJ, USA) and blood for serum isolation was collected in serum tubes (BD Biosciences, Franklin Lakes, NJ, USA). Whole blood was centrifuged at 600× *g* to remove the plasma and buffy coat, and red blood cells were washed thrice with RPMI 1640 with Glutamax, supplemented with 0.1% glucose, 30 mM HEPES, and 200 µM hypoxanthine (*Pcy*-incomplete media) and stored at 4 °C. A donor screen was carried out with 16 different NHPs to identify a serum lot supporting *Pcy* in vitro culturing. Serum was collected, heat inactivated at 50 °C for 1 h with inversions every 15 min, filter sterilized, aliquoted, and stored in −20 °C. Donors were identified and booked for routine serum collection. The serum supporting the best *Pcy* growth was selected for in vitro culture and was added to *Pcy*-incomplete media at 20% *v*/*v* to create *Pcy*-complete media.

### 2.2. In Vitro Culturing of Pcy Berok K2 Parasites

Frozen vials of *Pcy* Berok K2 parasites were received from NITD Singapore [47]. Cryopreserved parasites were thawed using a deglycerolization protocol consisting of three sodium chloride solutions (12%, 1.2%, and 0.9%) and maintained in *Pcy*-complete media in reduced oxygen (5% oxygen, 5% carbon dioxide, and remaining nitrogen) incubators at 37 °C. Regular media changes were performed, and fresh *M. mulatta* red blood cells (RBCs) were supplemented in culture as needed to maintain 3–5% hematocrit.

### 2.3. Parasite Synchronization

*Pcy* Berok K2 culture was centrifuged at 600× *g* for 5 min to obtain a packed parasite culture. The pellet was incubated at room temperature for 20 min with 40× volume of sterile filtered 140 mM guanidine hydrochloride solution prepared in 20 mM HEPES buffer. The mixture was then washed by centrifuging at 600× *g* for 5 min followed by washing twice with *Pcy*-incomplete media. The packed pellet was resuspended into media at 3–5% hematocrit and monitored for 48 h until the majority of the parasites in the culture reached the schizont stage (within 40 to 48 h window). To enrich for mature schizonts from the culture, density gradient purification was performed using Nycodenz (Progen, Heidelberg, Germany) as previously described [49]. In brief, the packed pellet was resuspended in 1 mL *Pcy*-incomplete media to 50% hematocrit which was carefully layered over 5 mL of 55% Nycodenz solution diluted with phosphate-buffered saline (PBS) in a 15 mL tube and centrifuged at 900× *g* for 12 min without brake. A thin layer of schizont pellet was observed in the middle top layer and was gently extracted using a sterile Pasteur pipette. Schizonts were washed twice with *Pcy*-incomplete media and used for downstream transfections.

### 2.4. The Generation of a Cas9 Plasmid for the Editing of P. cynomolgi

The plasmid pYC-L2 was used as the backbone [50,51]. The following components were amplified from *Pcy* Berok K2 genomic DNA and ligated into the plasmid, replacing the *P. yoelii* components to produce the pCC-L2 plasmid: EF1a bidirectional promoter, U6 promoter, and HSP70 terminator (Appendix A). The CSP cassette was ligated into this pCC-L2 plasmid. See Figure 1A for plasmid map.

### 2.5. The Generation of a Chimeric Pv CSP Cassette to Replace Pcy Berok CSP

To generate a plasmid targeting the CSP region of *Pcy* Berok K2, a 3033 bp region was first amplified via polymerase chain reaction (PCR) from a *Pcy* Berok K2 in vitro pellet following methods in 2.7 below for genotyping and Sanger sequenced by a commercial supplier, based on the available *Pcy* M strain reference [52,53] (primer sequences in Appendix A). The chimeric *Pv* sequence was based on a previously published sequence [54] with the plasmid kindly provided by Blandine Franke-Fayard and Chris Janse. In brief, this chimeric CSP includes the conserved N- and C-terminal regions of the protein with a chimeric repeat region consisting of equal stretches of the VK210 and VK247, in this order. An additional methionine after the signal peptide cleavage site in this sequence was removed with a fusion PCR to generate the final sequence encoding the 294aa *Pv* CSP chimeric repeat protein (Figure 1A, also see Miyazaki [54]).

Based on the sequence data we had generated for the CSP region of *Pcy* Berok, 700 and 685 bp regions of the 5′ and 3′ UTR, respectively, were amplified from genomic DNA and fused with the chimeric *Pv* CSP using primer overhangs in a fusion PCR. This CSP cassette was inserted into the pCC-L2 plasmid with SalI and NotI restriction sites (Appendix A). Using the online program CHOPCHOP (see Labun et al. [55]), three guide sequences were selected and inserted into the pCC-L2-PvCSP plasmid downstream of the U6 promoter with the Esp3I restriction enzyme to generate pCC-PvChimera-CSP-G2/13/16 plasmids. Due to undesirable mutations identified in the CSP region of the G2 and G16 plasmids, plasmid G13 was used for transfections (named Pcy_Pv plasmid, below).

### 2.6. Nucleofection of Pcy Berok K2 Parasites

The protocol to transfect *Pcy* schizonts was adapted from tools used to genetically modify *P. knowlesi* [56]. Nucleofections were performed using Lonza Amaxa 4D electroporator and P3 Primary cell 4D Nucleofector X Kit L (Lonza, Walkersville, MD, USA). A volume of 5–10 µL of purified *Pcy* schizonts was incubated with 5 µL of 20 µg Pcy_Pv CSP plasmid and 90 µL of P3 primary cell solution for 5 min. The mix was transferred to a 100 µL single nucleocuvette and immediately electroporated using program FP 138. Electroporated parasites were transferred to a 1.4 mL Eppendorf tube containing 500 μL prewarmed *Pcy*-complete medium supplemented with 25 μL fresh packed RBCs and incubated at 37 °C on a thermomixer, shaking at 650 rpm for 30 min. Transfected parasites were transferred to 24-well plate containing 1 mL *Pcy*-complete media and incubated as described above.

Media change was performed 24 h after transfection and selection with 50 nM pyrimethamine was initiated; media was changed daily until the culture was free of parasites. Parasites were below limit of detection by Giemsa smears by day 7 of selection, after which selection was stopped. *Pcy*-complete medium was used for maintenance with media change performed 3× weekly; 20% of total culture volume was cut and replaced with fresh blood and media weekly, and periodic Giemsa smears were assessed to monitor culture health and detect parasites.

### 2.7. Genotyping and Sequencing of Pcy Transgenic Parasites

Sequencing of transgenic *Pcy*[*Pv*CSP] parasites from in vitro culture was set up using RBCs from culture and thermocycling conducted with Phusion Blood Direct kit (Thermo Fisher, Waltham, MA, USA) to confirm recombinant locus. Sequencing of blood-stage *Pcy*[*Pv*CSP] parasites from in vivo NHP infections was conducted using parasite pellet from in vitro cultures set up with NHP blood and genomic DNA isolated using the DNeasy blood and tissue kit from Qiagen (Germantown, MD, USA), following manufacturer’s protocol. Three sets of primers were used for the PCR. The first set (P1-CATATCTGTACATGTCCATGTAGTGACC and P3-GAATACTACTCACGGCGAGC) was used to amplify a 786 bp fragment at the 5′ end of the recombinant locus *Pcy*[*Pv*CSP], and second set (P4-TATTCTGCTGGTGCTGGCCC and P2-ACGCAGTTTGCACACACCTGGC) was designed to amplify a 715 bp fragment at the 3′ end of the *Pcy*[*Pv*CSP] recombinant locus. The last set (P6-GTAAGAATGAGAAGAAAAGTTAGTGC and P2-ACGCAGTTTGCACACACCTGGC) was designed to amplify a 1134 bp fragment from the wild-type 3′ CSP locus. All three cultures, *Pcy*[*Pv*CSP] 1, *Pcy*[*Pv*CSP] 2, and *Pcy*[*Pv*CSP] 3, along with *Pcy* Berok K2 wild-type parasites and plasmid DNA as controls were included in the PCR diagnostic.

### 2.8. Mosquito Infections

*Anopheles stephensi* mosquitoes were reared at OHSU or purchased from the University of Georgia SporoCore facility (Athens, GA, USA). Female mosquitoes were maintained in an incubator at 26 °C, 80% humidity, and provided a sugar cube and water pad until the day of infection. Three-to-six-day old female mosquitoes were infected via a membrane feeder with blood drawn into a Lithium-Heparin vacutainer (Greiner Bio-One, Monroe, NC, USA) from an infected RM. To prevent exflagellation of the male gametes prior to uptake by mosquitoes, the blood was maintained at 39 °C during transport from the NHP facility to the insectary. To minimize the potential contribution of anti-*Pcy* antibodies, for some infections the RBCs were washed twice with RPMI and sera replaced with human AB+ sera. Mosquitoes were allowed to feed until all were either visibly engorged with blood (typically 75–90% of all mosquitoes) or lost interest. Infected mosquitoes were maintained at 26 °C, 80% humidity, and provided a sugar cube and water pad, changed daily. At 4–6 days post-infection, mosquitoes were provided a supplemental blood meal via an anesthetized mouse [57]. At 8–10 days post-infection, a sample of mosquitoes were dissected and midguts visualized under phase-contrast microscopy to quantify oocyst burden. At D16–19, mosquitoes were dissected and sporozoites quantified for downstream use, including infection of a naïve NHP, immunofluorescence assays, and cryopreservation.

### 2.9. Immunofluorescence Assay Demonstration of Sporozoite Specificity for P. cynomolgi CSP Antibodies

Freshly dissected sporozoites were enumerated, fixed in 4% paraformaldehyde for 20 min at room temperature, and, following a PBS wash, were loaded on a multiwell microscope slide at 25,000 sporozoites per 30 μL. Slides were allowed to air dry overnight, wrapped in aluminum foil, and stored at −80 °C until permeabilization. After thawing, sporozoites were washed twice with 1× PBS and then permeabilized and blocked with 3% *w*/*v* bovine serum albumin plus 0.2% *v*/*v* Triton X-100 in 1× PBS for 1 hour at room temperature. Primary mouse mAbs to *P. vivax* CSP (Pv-247-CDC and Pv-210-CDC) were obtained through BEI Resources (NIAID, NIH: *Plasmodium vivax* Sporozoite ELISA Reagent Kit, MRA-1028K, contributed by Robert A. Wirtz) and diluted in 1× PBS/3% BSA and incubated overnight at 4 °C in a humidified chamber. Following two washes with 1× PBS, fluorescent secondary antibodies were diluted in 1× PBS/3% BSA and incubated for 1 h at room temperature in a humidified chamber and shielded from light. Nucleic acid was then stained with DAPI in 1× PBS for 10 min at room temperature. Sporozoites were washed three times with 1× PBS, and then slides were mounted with one drop of ProLong Antifade reagent (ThermoFisher, Waltham, MA, USA) and topped with a cover slip. After 30 min, slides were sealed with nail polish and stored at 4 °C until imaging. Images were acquired using Olympus (Bartlett, TN, USA) 1x70 Delta Vision deconvolution microscopy.

### 2.10. Infection of NHPs

Blood-stage parasites preserved in glycerolyte were stored in liquid nitrogen until use and fresh sporozoites were used shortly after dissection. For infections initiated by asexual blood-stage parasites, a 200–500 μL parasite stock originating from culture or previous RM infection was thawed and deglycerolization performed using saline washes as described [58]. For infections initiated with sporozoites, 36,000 fresh sporozoites were washed once in sterile PBS prior to infection. For all types of infections, the parasite pellet was suspended in 1 mL sterile PBS and injected intravenously via the saphenous or cephalic vein. Beginning 5–8 days post-infection, a small blood sample was collected for detection of parasites via thin-smear microscopy. Complete blood counts were collected every 1–3 days during infection to monitor hematocrit and hematological factors associated with animal well-being. During peak infection, larger draws of several milliliters were collected to feed to mosquitoes and preserve parasites for validation of *Pv*CSP transgene integration and production of stocks for future infections. Animals were treated prior to a maximum parasitemia cutoff of 300,000 parasites/μL. To clear blood-stage parasites, Coartem (20 mg artemether/120 mg lumefantrine, Novartis) was administered orally in a treatment course of 1 tablet twice daily for three consecutive days. For animals infected with sporozoites, semi-weekly monitoring was conducted to detect relapse infection from a reactivated hypnozoite. At the conclusion of the study, sporozoite-infected animals were treated with a single dose of Tafenoquine (150 mg, GlaxoSmithKline, London, UK; trade name Krintafel) to clear any remaining hypnozoites in addition to a course of Coartem.

### 2.11. Quantification of Parasites in NHP Blood

Thin-smear microscopy to enumerate parasitemia was performed daily during patent infection and twice weekly during relapse monitoring to track the course of infection by Giemsa-staining and quantification of >20,000 RBCs. To confirm the absence of parasites in the blood between blood-stage treatment and relapse, blood samples were preserved in RNA*later* solution (Invitrogen, Waltham, MA, USA) and parasites quantified by qRT-PCR. In brief, extraction of nucleic acid from blood samples preserved in RNA*later* solution (200 μL blood in 400 μL RNA*later*) was performed using the MagMAX^TM^-96 Total RNA Isolation Kit (Invitrogen). This was followed by pipetting of mastermix and template into 96-well plates for one-step amplification on the StepOnePlus™ instrument (Applied Biosystems, Waltham, MA, USA). A Pan-Plasmodium 18S rRNA assay (Forward PanDDT1043F19: 5′-AAAGTTAAGGGAGTGAAGA-3′; Reverse PanDDT1197R22: 5′-AAGACTTTGATTTCTCATAAGG-3′; Probe: 5′-[CAL Fluor Orange 560]-ACCGTCGTAATCTTAACCATAAACTA[T(Black Hole Quencher-1)]GCCGACTAG-3′[Spacer C3]) alongside a housekeeping gene (TATA binding protein; Forward: 5′-GATAAGAGAGCCACGAACCAC-3′; Reverse: 5′-CAAGAACTTAGCTGGAAAACCC-3′; Probe: 5′-[FAM]-CACAGGAGC-[ZEN]-CAAGAGTGAAGAACAGT-[3IABkFQ]-3′) was used for quantification using the SensiFAST™ Probe Lo-ROX One-Step Kit (Bioline, London, UK). Final concentrations of 0.08 μM for each probe and 0.64 μM for all primers were used for both targets, in a 25 μL reaction volume. Cycling conditions consisted of reverse transcription (10 min) at 48 °C, denaturation (2 min) at 95 °C, and 40 PCR cycles of 95 °C (5 s) and 50 °C (35 s). Samples were run in triplicate and compared to a standard curve (armored RNA) for quantification.

## 3. Results

### 3.1. The Generation of a P. cynomolgi Parasite Expressing the P. vivax Circumsporozoite Protein

A single plasmid carrying the *Pv* CSP chimera insert, *Cas*9 endonuclease, and guide sequence was used to target the endogenous *Pcy csp* locus (Figure 1A). Of the 15 transfection reactions performed, only three cultures (“T1–T3” in Figure 1B) had parasites emerge 30 days post-nucleofection. Culture T1 showed no presence of wild-type parasites by the PCR, whereas Culture T2 had only wild-type parasites. The PCR of Culture T3 indicated a mixed population of wild-type and transgenic parasites (Figure 1B). The selection marker, human dihydrofolate reductase, is not integrated in the locus but could be retained episomally. To test for episomal expression, Culture T1 was treated with 50 nM pyrimethamine. Parasite death was observed with culture clearance within 1 week which, combined with the PCR results in Figure 1B, indicate no or an extremely low presence of episomal plasmid. Parasites from this culture were used for all in vivo experiments. In summary, *Pcy* Berok K2 wild-type parasites were successfully modified to generate transgenic parasites expressing the *Pv* circumsporozoite gene integrated in chromosomal DNA using *Cas9*-mediated transgenesis. This parasite is referred to as “*Pcy*[*Pv*CSP]”.

### 3.2. The Pcy[PvCSP] Transgenic Parasite Completes the Life Cycle Between Rhesus Macaques and Anopheles Mosquitoes

We set out to determine if the *Pcy*[*Pv*CSP] transgenic parasite infects macaques at the blood stage, if it can develop in the mosquito vector to form infectious sporozoites, and if these sporozoites are infectious in vivo with the ability to form relapsing hypnozoites (overview in Figure 2A). A cryopreserved stock of the *Pcy*[*Pv*CSP] transgenic parasite produced in vitro was used to initiate an infection at the blood stage in RM1 for the purpose of parasite expansion and the generation of cryopreserved blood stocks. After a prepatent period of 7 days, the peak parasitemia was 37,927 parasites/μL (0.85%) at D11, which began to decline at D12 and continued to decline after treatment at D13 according to the planned protocol (i.e., not treated due to clinical treatment endpoint) (Figure 2B, leftmost panel). This NHP-cycled parasite was used to initiate an infection in RM2 to test the capacity of the infected blood to infect mosquitoes and generate sporozoites. Parasitemia in RM2 was first identified on D8 post-infection, increasing to 77,905 parasites/μL until the administration of the antimalarial treatment on D12 (Figure 2B, second panel). This course of infection is similar to that seen in our lab for the *Pcy* Berok strain wild-type (WT) parasite. On days 11 and 12 post-infection, blood was fed to *A. stephensi* mosquitoes via a membrane feeder. A small number of mosquitoes were dissected for the detection of midgut oocysts at D10 post-blood meal. Oocysts were not detected in any mosquitoes dissected from the D11 feed, and 6/10 mosquitoes contained at least one oocyst in the mosquitoes from the D12 feed. Mosquito salivary glands from the D12 feed were dissected at D16 post-blood meal which yielded 11,930 sporozoites/mosquito. We next intravenously injected 36,000 of these sporozoites into RM3 and monitored for primary and relapse infection (Figure 2B, third panel). Parasites were detected by microscopy starting on D11 post-infection and increased exponentially starting at D16. Mosquito feeds were conducted on days 19, 20, and 21 as the parasitemia approached 100,000 parasites/μL. A blood-stage treatment with Coartem was administered on D23, and the animal became microscopy negative at D27. Importantly, the administration of Coartem does not eliminate hypnozoites, and so the semi-weekly monitoring of parasitemia by thin-smear microscopy was initiated to monitor for hypnozoite reactivation after the clearance of the primary infection. Parasites were again detected by microscopy on D46 (19 days after becoming microscopy negative post-Coartem), suggesting that the sporozoites maintained the ability to establish relapsing infections. The relapse infection peaked at a lower parasitemia of 31,571 parasites/μL on D53 before slowly self-clearing, and Coartem was administered on D59 to ensure the complete clearance of blood-stage parasites. A single mosquito feed performed on the relapse infection yielded minimally infected mosquitoes (702 sporozoites/mosquito) but confirms the ability of relapse parasites to infect the mosquito vector. On D125 post-infection, a radical cure treatment was administered to clear any potential remaining hypnozoites. A quantitative reverse transcriptase PCR (qRT-PCR) was used to provide a more sensitive measure of parasitemia, particularly between D28 and D39 to determine if we observed a true relapse rather than the recrudescence of sub-microscopic parasite infections (Appendix A). Indeed, we found the qRT-PCR to be more sensitive than microscopy, with parasitemia persisting until D28 (vs. D27 via microscopy) but with a distinct return to baseline on D35. These results indicate a relapse, rather than recrudescence of a sub-microscopic infection.

### 3.3. Transgenic Parasites Maintain Transgene Integrity and Protein Expression In Vivo

We next set out to ensure that we maintained the transgene integration through the three RMs using blood from the primary and relapse infections of RM3. The transgene integration was confirmed by a PCR using primers specific for the 3′ and 5′ ends of *Pv* and *Pcy* CSP on DNA isolated from parasite-enriched parental (*Pcy* Berok K2) or RM-passaged transgenic (*Pcy*[*Pv*CSP]) blood. The positive bands for the *Pcy*[*Pv*CSP] parasite using *Pv*CSP primers, and the lack of bands using parental *Pcy* primers, indicates the successful integration and complete replacement of the parental gene (Figure 3A). The transgenic *Pv*CSP contained the two serotypes, VK210 and VK247, that have been found across malaria-endemic areas and can co-exist in the same region and same patients [38] and are highly heterogenous [59,60]. Thus, our transgenic parasite would have maximum utility only if it expressed both variants of CSP at the protein level to enable the preclinical assessment of anti-*Pv*CSP interventions. The expression of the chimeric *Pv*CSP was tested by an immunofluorescent assay (IFA) using mAbs that bind to the VK210 (clone 2F2) and VK247 (clone 2E10) variants. In contrast to the field isolates of sporozoites from Thailand and Peru that were only recognized by the anti-VK210 antibody, the *Pcy*[*Pv*CSP] sporozoites were recognized by both mAbs (Figure 3B). Together, these data indicate that the *Pcy*[*Pv*CSP] parasite expresses both *Pv*CSP serotypes and maintains the in vivo capacity to complete the parasite life cycle, including relapse, in NHPs. In addition to the first gene replacement of the *Plasmodium cynomolgi* chromosomal DNA (vs. the previous use of a maintained artificial chromosome [61,62] or point mutation [48]), this parasite will enable the first in vivo assessment of anti-*Pv*CSP interventions to protect against primary and relapse infections in immune-competent animals via a stringent parasite challenge.

## 4. Discussion

In contrast to other *Plasmodium* species for which transgenesis has been commonplace, including for the non-relapsing NHP parasite *P. knowlesi* [36,37,38,39,40,41,42,43,44,63,64,65], the transgenesis of the closely related relapsing parasites *P. vivax* and *P. cynomolgi* has proven more difficult due to the lack of an in vitro culture. This work capitalized on the recent in vitro adaptation of *Pcy* which enabled a *Cas9*-mediated transgenesis without the need for parasite cycling and selection in vivo [47,48]. We used this opportunity to address the practical aim of a parasite that can be used to test CSP-based *Pv* interventions against primary and relapse infections. To this end, we have demonstrated the ability to perform the parasite transgenesis of *Pcy* and the capacity for our *Pv*CSP-expressing *Pcy* transgenic parasite to successfully complete the entire parasite lifecycle, including the infection of RMs at both the blood and liver stages, the full development in the mosquito vector, and the formation of hypnozoites that reactivate to generate relapse infections. The circumsporozoite protein serves several critical functions in the mammalian host and in the mosquito vector [11]. While several CSP mutant parasites have been generated in *P. berghei* [66,67], the necessity for the correct function of this protein in sporozoite development is emphasized by an attempt to manipulate CSP in *P. falciparum* that resulted in parasites that abrogate at the oocyst stage [68]. Thus, the capacity of our transgenic parasite to not only complete development in the mosquito but also infect the NHP liver is encouraging even if unsurprising given the close phylogenetic relationship and overlapping hosts of *Pv* and *Pcy*. To our knowledge, this is the first demonstration of gene replacement in a *relapsing* parasite and the only preclinical means of assessing the efficacy of anti-*Pv* CSP interventions in a relapsing model.

The steps in the formation and activation of hypnozoites have recently been identified as a key knowledge gap in understanding *Pv* biology and thus in driving its control and elimination [69]. This gap is perpetuated by the difficulty in obtaining *Pv* parasites, which cannot be maintained in long-term cultures. If *Pv* sporozoites are obtained, they can be used to infect cultured hepatocytes to observe dormant and replicating forms and have recently been used in humanized liver mice to demonstrate not only primary and relapse infections but the susceptibility to mAbs [70]. Still, the combination of logistical hurdles, difficulties in perpetuating the blood-stage infection, and the absence of the immune system either in vitro or in humanized liver mice clouds our understanding of complex host–pathogen interactions. Previous studies have used the *Pcy* model to not only study the biology of infection, but to generate parasites carrying additional genes that allow for quantification and visualization in vitro [61]. As mentioned above, these studies utilized extra-chromosomal DNA to introduce new genes rather than the knockout, knockin, or mutation of genes possible with the *Cas9*-based method used here. A more recent study used CRISPR-based mutations to knock-in a point mutation related to drug resistance [48]. We hope our results are an important step to the routine transgenesis of *Pcy* parasites that enables a greater biological understanding of relapsing malaria.

In addition to a contribution to the field of parasite transgenesis by the generation of transgenic relapsing parasites, these studies also have the potential to advance vaccine and mAb development for *Pv* though improved challenge capacities. While valuable advancements in *Pv* vaccine testing have been made using transgenic murine parasites expressing the *Pv* CSP [71], these rodent parasites do not have the capacity to form hypnozoites and thus are not well suited to test the effect of vaccine candidates on reducing the hypnozoite reservoir. The reactivation of hypnozoites is proposed to cause ~80% of the observed *Pv* infections [72]. This means that interventions that can reduce sporozoite infections of the liver, even if unable to completely abrogate infection, may have an outsized effect on *Pv* elimination. For example, while a vaccine or mAb prophylactic capable of stopping 90–95% of *Pf* sporozoites from reaching the liver still results in a blood-stage infection 50% of the time [73], modeling indicates that the same reduction in the liver burden for a *Pv* vaccine could reduce total blood-stage infections by up to 90% [73] and thus have an outsized impact on disease and transmission compared to *Pf*. As mentioned above, a humanized liver mouse model assessed the effect of passively transferred mAb to prevent primary and relapse infections [26], but this model is not suited to testing active immunization or interactions between antibodies and FcR-bearing cells as the humanized liver mice are severely immunocompromised. We hope our parasite can be used to test the effect of novel interventions in the context of a more complete representation of *Pv* biology.

While this transgenic parasite serves as an important step in the study of relapsing *Plasmodium* infections, there are several limitations. First, only the CSP gene of *Pcy* is replaced with the *Pv* homolog, and thus it is only suitable for testing CSP-based interventions. The CSP is particularly suited to transgenesis given the lack of conservation between key antibody epitopes between *Pcy* and *Pv* with likely the maintenance of function. There is growing consensus in the malaria vaccine community that a multi-antigen and multi-stage approach to vaccination will be required for long-term control or elimination with a vaccine or mAb. As such, developing further transgenic parasites to expand the number of vaccine targets through repeated transgenesis would be useful. However, the close homology between *Pv* and *Pcy* can enable the testing of vaccines against *Pcy* proteins that likely share a conserved function, as performed for SSP2/TRAP [74], or by better defining the direct sequence homology between the target proteins in *Pv* and *Pcy* to assure the suitability of the *Pcy* parasite for relevant *Pv* vaccine testing as recently achieved with *Pv* AMA1 [75]. Nonetheless, this study serves as an important proof-of-concept demonstrating that the *Pcy* parasite can be genetically manipulated to create chimeras expressing *Pv* proteins. In addition, while we demonstrate clearly that the *Pv* CSP is expressed on our *Pcy*[*Pv*CSP] parasite and the *Pcy* CSP gene was absent, we were unable to confirm at the protein level that the *Pcy* CSP was not co-expressed as we do not have an mAb which reacts against only *Pcy* CSP.

In summary, studying the liver stages of *Pv* is extremely challenging and these challenges drastically slow the progress of *Pv* interventions. The expanded capabilities of animal models relevant for understanding *Pv* biology and testing interventions will be key for advancing interventions against *Pv*. This transgenic parasite provides a useful addition to the limited research tools available to study relapsing *Plasmodium* biology and to develop interventions against this species of malaria.

## 5. Conclusions

In this work we aimed to develop a new tool for testing *Pv* CSP-based pre-erythrocytic interventions. We demonstrate that the *Pcy*[*Pv*CSP] transgenic parasite we generated expresses the *Pv* CSP in place of the native *Pc* CSP, is infectious to both *A. stephensi* mosquitoes and rhesus macaques, and retains the ability to form dormant hypnozoites that cause relapse. This parasite is a valuable addition to the available tools for testing interventions against *Pv*.

## Figures and Tables

**Figure 1 vaccines-13-00536-f001:**
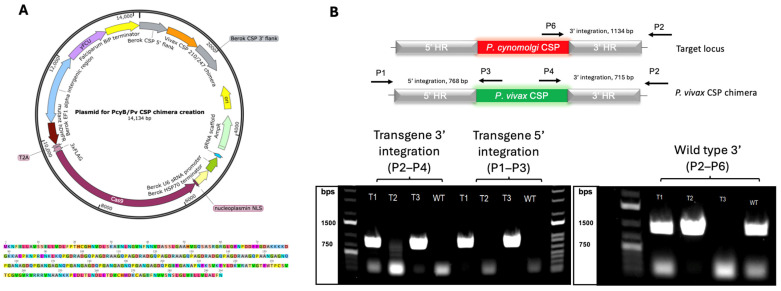
The construction and validation of a *P. vivax*-expressing *P. cynomolgi* parasite. (**A**) The plasmid map and CSP sequence used for the Cas9-mediated generation of the *Pcy*[*Pv*CSP] transgenic parasite. (**B**) PCR results confirming the successful integration of the *Pv* csp gene replacing the native *Pcy* csp after the in vitro expansion of the transfected parasites. The top graphic indicates the primers and expected amplicon size with genotyping primer pairs. Bottom images show PCR gels of transfected cultures The T1–T3 using the primers indicated above. Culture T1 is shown to have the wild-type *Pcy* CSP (band at ~1200 pb on right image) as well as the *Pv* CSP (bands at ~750 pb on left image for both 3′ and 5′ integration sites), indicating a mixed population. Culture T2 has only wild-type parasites, evidenced by the lack of a band at ~750 bp for either transgene integration site (left gel). Culture T3 is a pure population of transgenic parasites, evidenced by the band at ~750 bp for both 3′ and 5′ *Pv*-transgene integration sites and the lack of a band at ~1200 bp for the *Pcy* wild-type CSP detection.

**Figure 2 vaccines-13-00536-f002:**
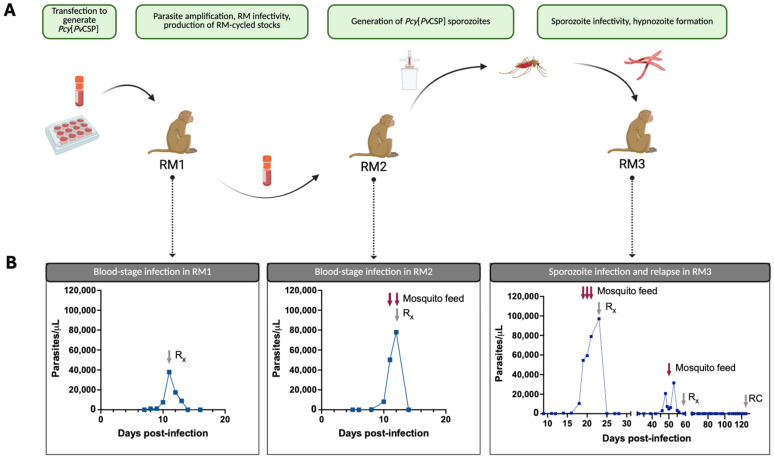
A *Pcy*[*Pv*CSP] transgenic parasite completes the parasite life cycle between rhesus macaques and *A. stephensi* mosquitoes including relapse. (**A**) An overview of the NHP infections and mosquito feeds, with the purpose of each stage indicated above. (**B**) Parasitemia curves demonstrating the infectivity of the transgenic parasite at both blood and sporozoite stages. The parasitemia presented was quantified by thin-smear microscopy, and parasite-negativity for animals infected with sporozoites (center panel) was confirmed by a qRT-PCR (see Appendix A). *Pcy*[*Pv*CSP] = *Plasmodium cynomolgi* parasite with an endogenous circumsporozoite gene replaced with a *P. vivax* sequence. RM = rhesus macaque. Rx = antimalarial treatment specific to clearing blood-stage parasites (no action against hypnozoites). RC = radical cure with Tafenoquine to clear hypnozoites.

**Figure 3 vaccines-13-00536-f003:**
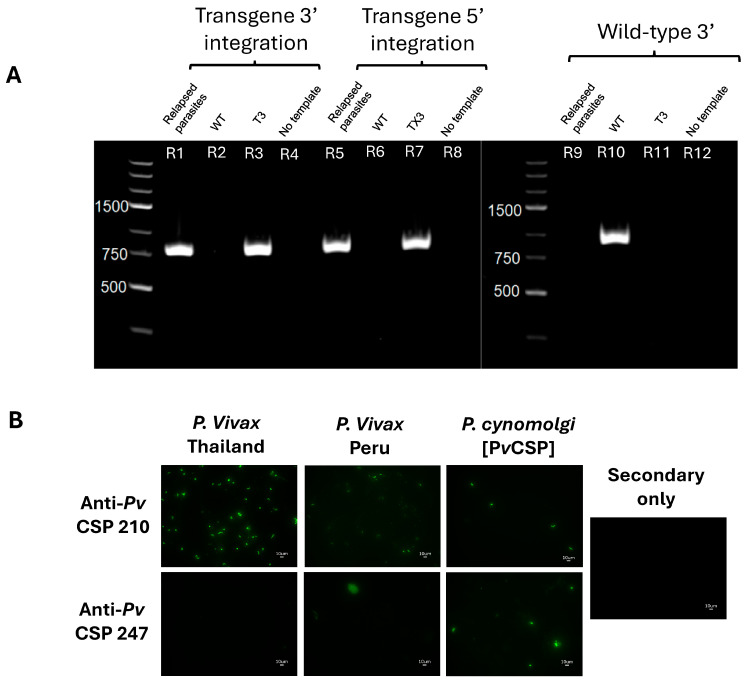
Confirmation of *Pv*CSP integration and protein expression following infection of NHPs and cycling in *A. stephensi* mosquitoes. (**A**) PCR for integration of *Pv* csp gene replacing native *Pcy* csp after cycling in three NHPs and *Anopheles* mosquitoes using infected blood samples from relapsed parasites, wild-type parental non-transgenic parasites (WT), Culture T3 transgenic parasites (T3), and no template sample using primer sets described in Figure 1. (**B**) Immunofluorescent imaging comparing expression of two major *Pv* CSP variants in parasite field isolates from Thailand and Peru, both of which predominantly express VK210, to our *Pcy*[*Pv*CSP] transgenic parasite that expresses both VK210 and VK247. Peruvian strain stained with anti-Pv CSP 247 exhibits one location of autofluorescent debris in upper left quadrant and is distinct in size and focus-plane from true sporozoites as no defined sporozoites could be found. Secondary only control lacks any CSP-specific antibody.

## Data Availability

The original contributions presented in this study are included in the article/Appendix A. Further inquiries can be directed to the corresponding author(s).

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
