# Peer review of "Generation of a Transgenic Plasmodium cynomolgi Parasite Expressing Plasmodium vivax Circumsporozoite Protein for Testing P. vivax CSP-Based Malaria Vaccines in Non-Human Primates"

_vaccines, 2025, doi:10.3390/vaccines13050536_

Round 1
Reviewer 1 Report
Comments and Suggestions for Authors
General comments:
In this paper the authors described for the first time an example of gene replacement in cultured P cynomolgi parasites through reverse genetics. They further ascertain that the transgenic parasite can be adapted to in vivo growth in rhesus macaques and transmitted to mosquitoes. In a nutshell, the authors described a method to make P cynomolgi transgenics in culture using the Cas-9 method. While this is interesting in its own right, the technological advancement is limited to the adaptation of Cas-9 mediated integration to P. cynomolgi. In fact, the re-adaptation of a transgenic non-human primate parasite (Pk CSP KO) to in vivo growth in rhesus macaques has been described before (Kocken at al 2002 Infection and Immunity), but is not referenced by the authors. Moreover, the success of said re-adaptation and transmission is heavily dependent on the type of transgenic parasite that has been used and is not necessarily generalisable. While the authors have discussed Chua et al 2019 in the context of the culture-adaptation of P. cynomolgi, they do not discuss the importance of the Chua et al 2019 paper in the context of the in vivo characterisation of the P. cynomolgi Berok strain in rhesus macaques. This is important to define the phenotypical characteristics of the strain in question particularly when discussing the implications of the use of said strain in Pv related vaccine research. The authors should also consider including references to Cas-9 mediated ortholog replacement in the non-human primate malaria P. knowlesi (Mertens et al Vaccine 2024 42(16): 3621) as a tool for vaccine research. Finally, I think it is important that the authors avoid overstating the novelty of a Cas-9 based approach which they have effectively adapted from other malaria parasites to use in P. cynomolgi rather than developing a new method.
While the paper is therefore important and should be published, some of the claims made in the discussion are exaggerated and not reflected by the data presented in the article. The ability to perform gene replacement in cultured P. cynomolgi does not immediately revolutionise the Plasmodium vivax vaccine research field as the authors claim in the discussion nor does it amount to the establishment of a new animal model.
The author themselves recognise the limitation of their claim making (p. 12 line 457-458) by acknowledging that p. 12 lines 458-460: “First, only the CSP gene of Pc is replaced with the Pv homolog and thus it is only suitable for testing CSP-based interventions.”
Moreover, the authors appear to think that approaches to vaccine research for Pv need necessarily to be based on the introduction of Pv genes in P cynomolgi, which is not necessarily the case. While this may be one approach, Pv-related vaccine research using the P. cynomolgi-rhesus macaque model based on the phylogenetic relatedness of the P cynomolgi parasite to P. vivax malaria and on the rhesus macaque immune system to the human one has been performed for a very long time. The authors lack of acknowledgement for this research is a problem as prior work should never be discounted.
To seriously bridge this research and the broader field of vaccine research, the authors should consider the limitations of the P cynomolgi Berok strain, which has been culture adapted, but presents a different phenotype (e.g. low number of in vivo hypnozoites in the liver) as described in the old literature by Collins, Barnwell et al for example J Parasitol. 1999 Apr;85(2):268-7; Sutton PL Infect Genet Evol. 2016 Jun;40:243-252, Chua et al 2019), which may not be ideal for the vaccine research. In this context, the authors themselves described the ability to form reactivating hypnozoites as key: “Reactivation of hypnozoites is proposed to cause ~80% of the observed Pv infections57.”
Thus overall, we suggest the publication of this article after major modifications of the discussion to better acknowledge the prior work in the field of Pv vaccine research using the P. cynomolgi-rhesus macaque model. We suggest that the authors avoid overstating the significance of the technological advances presented and engage in a discussion of the advantages (in vitro culture) and disadvantages (phenotype) of the Berok strain in the context of Pv vaccine research. Furthermore, we invite the authors to conduct minor revisions on the introduction to clarify specific areas of their writing.
Detailed comments:
Abstract:
p.1 line 17: “Pf is only one of the four major species of Plasmodium that infect humans”
This statement is factually not correct. At the moment there are at least 6 Plasmodium species that infect humans some of which are zoonotic.
Introduction:
p. 2 line 63-64: “Indeed, the only licensed malaria vaccines to-date, RTS,S/AS01 (MosquirixTM) and R21/Matrix-MTM, both act at the pre-erythrocytic stage”
While both vaccines contain the major pre-erythrocytic antigen CSP, the mechanism of action is not yet clear. The fact that these vaccines reduce severe malaria cases but do not prevent infection with malaria has led to uncertainty on whether these vaccines indeed act at the pre-erythrocytic stage. This statement should be nuanced.
p. 3 line 94-95: “Thus, Pc has been a cornerstone of Pv research25,31,32 and indeed, hypnozoites were first described for Pc35 before their identification in Pv (reviewed in 27)”
The abbreviation Pc is misleading. Pc is normally the abbreviation for the rodent malaria parasite P. chabaudi. To distinguish it from P. chabaudi, P. cyonomlgi has normally been termed Pcy. Please revise accordingly.
p. 3 line 106-107: “However, these in vitro parasites cannot yet be directly transmitted to mosquitoes as is commonplace for Pf.”
Please explain why this cannot be done in culture P. cynomolgi K2. Also, please nuance this statement as Pf in long term culture (e.g. most 3D7 lines) have also lost the ability to transmit.
p. 3 line 161: “based on the available Pc M strain reference (PlasmoDB)”
Please provide reference to the publication in which the Pcy M strain reference genome is described (Pasini et al. Wellcome Open Res. 2017:2:42; Tachibana et al. Nature Genetics 2012: 44, 1051)
p. 5 line 222-223: “At 4-6 days post-infection, mosquitoes were provided a supplemental blood meal via an anesthetized mouse”
Please explain this choice. Why was blood from mouse used instead of blood from rhesus macaque for the supplemental blood meal? This leads to the mosquitoes taking up blood from two different animals.
p. 4 line 162: “ The chimeric Pv sequence was based on the previously published sequences with the plasmid kindly provided by…”
p. 7 line 294: “ Pv CSP chimera insert”
Please define what the chimera is. A reference to a publication is not sufficient as it is argued that the chimeric Pv sequences is based on but not equivalent to the one described in the publication. The reader finds out later that the chimera is based on two CSP serotypes, but it is unclear at a molecular level how they differ and how they are combined in the chimera.
p.7 line 295-296: “Multiple transfection reactions were carried out…”
The only previous cited example of Cas-9 mediated gene editing in P cynomolgi (ward K et al JID 2023) described low transfection efficiency and low success rate. It is therefore important for the community to understand the transfection success rate that has been achieved here both in the context of reproducibility and general utility of the system described – especially compared to the more advanced P. knowlesi transgenesis system.
p. 7 line 299-303: “The selection marker, human dihydrofolate reductase, is not integrated in the locus but could be retained episomally. To confirm absence of episomal expression, Culture T1 was treated with 50 nM pyrimethamine. Parasite death was observed with culture clearance within 1 week indicating no presence of episomal plasmid.”
Without the use of a negative selection marker in pYC-L2 when pyrimethamine selection was relaxed a tiny proportion could keep this plasmid. To show that this has not happened, one would have to apply drug pressure to the transgenic parasite line for well over a week without parasites re-appering to conclude that the episome has indeed been lost. This is exemplified by how long it takes for transgenic parasites to develop after episomal transfection. Negative selection could have been carried out through the use of the yFCU cassette that perhaps the authors removed from the original plasmid.
Fig 1 portraying the plasmid and the insert is not readable nor annotated. This is a big problem as it is crucial to understanding the methodology used.
p 9 line 377-378: “The transgenic PvCSP contained the two 377 serotypes, VK210 and VK247, that have been found across malaria-endemic areas and can 378 co-exist in the same region and same patients”
Please explain the difference between the serotypes described in detail and reference Rosenberg R et al . Science 1989; Kane at al. Am J Trop Med Hyg 1993, 49, 478).
p 10 line 387-388: “In addition to the first gene replacement of Plasmodium chromosomal DNA…”
This is definitively a mistake. This is by no means the first gene replacement of Plasmodium chromosomal DNA. The statement should read: “This is the first gene replacement of Plasmodium cynomolgi chromosomal DNA…”
p. 10 Figure 3A: the P1-P12 subheadings are confusing as earlier primers are referred to by the same abbreviations.
p 10: Figure 3B the IFA identifies a large structure using anti-Pv CSP 247 in the P. vivax Peru strain, though the authors argue “In contrast to field isolates of sporozoites from Thailand and Peru that were only recognized by the anti-VK210 antibody, the 4 Pc[PvCSP] sporozoites were recognized by both mAbs (Figure 3B).”
However, the authors do not explain their finding. The staining rather appears to contradict what the authors are trying to prove. If this is autofluorescence this should be explicitly stated. Furthermore, a scale bar should be provided to go with the IFA pictures.
p. 11 line 405: “which enabled CRISPR-mediated transgenesis”
It is more appropriate to define it as Cas-9 transgenesis as you technically did not use the CRISPR system.
p.11 line 433-435: “As mentioned above, these studies utilized extra-chromosomal DNA to introduce new genes rather than the knockout, knock-in or mutation of genes possible with our CRISPR-based method.”
As the authors did not invent the CRISPR-based methodologies, which are now used for transgenesis in a variety of cells and microorganisms, and since technically they have used the Cas-9 based methodology rather than the CRISPR system it would be more fitting to avoid the pronoun OUR and to simply say: “possible with the Cas-9-based method described in this paper”
p.11 439-440: “In addition to a novel means to generate transgenic relapsing parasites, these studies will also advance vaccine and mAb development for Pv.”
How? It is unclear how this study specifically would advance Pv vaccine studies.
In this paper, the authors present a technological advancement, namely the application of a powerful molecular biology technique published in the 2010s (Cas-9) and its adaptation for use in P. cynomolgi. However, they do not show any vaccine (no data on immunisation, protection after challenge, immune responses) or mAb related data. Possible downstream effects on vaccine research have thus not been proven in this article and should thus not be claimed.
The way the authors present their contribution neglects all the previous vaccine research that has been done using the P cynomolgi-rhesus macaque model based on the phylogenetic relatedness of the P cynomolgi parasite to P. vivax malaria and on the rhesus macaque immune system to the human one. These include whole parasite vaccine studies (Pasini et al 2022 NPJ vaccines) and studies using adjuvated subunit vaccines (Kocken 1999, Dutta 2005, Kaushal 2007, Bardwaj 2002 and 2003, Cao 2000, Li 2004, Zhong 2000). Neglecting other people’s work with the P. cynomolgi-rhesus macaque model as a proxy for P vivax in the context of vaccine research, while discussing the limitations of the rodent models for Pv vaccine research is not a fair strategy to present as a game changer for the whole field of Pv vaccine development what is essentially a technological adaptation of an existing methodology that opens new possibilities.
While the adaptation of existing technology presented in this paper is surely interesting and is an interesting addition to the toolbox of technologies available for the study of parasite biology in vivo and in vitro through gene knock-out and knock-ins, the data presented here is not enough to claim an impact on P. vivax vaccine research as a whole. This also considering the limitations that the authors themselves recognise: p. 12 lines 458-460: “First, only the CSP gene of Pc is replaced with the Pv homolog and thus it is only suitable for testing CSP-based interventions.”
“This means interventions that can reduce sporozoite infections of the liver, even if unable to completely abrogate infection, may have an outsized effect on Pv elimination. For example, while a vaccine or mAb prophylactic capable of stopping 90-95% of Pf sporozoites from reaching the liver still results in a blood stage infection 50% of the time58”
This is an unfinished sentence (suggest removing the “while”) and what the authors are exactly trying to argue is unclear. The question of the efficacy of pre-erythrocytic vaccines does not hinge necessarily on preventing the sporozoite traveling to the liver. In fact, recent advances in whole parasite vaccines for Pf (Rozen et al Nat. Med 2025 31(1):218-222) with genetically attenuated lines have shown no blood stage infection or escape from the liver and a potent protection from challenge of over 95% with a single immunisation.
p 12 Lines 463-464: “or better defining the homology between the target proteins in Pv and Pc as recently done with Pv AMA159 will be critical”
This is unclear as the authors neglect all the work that has been done on Pv AMA1 including most recent work on diversity covering (DiCo) PvAMA1 vaccines aimed at by-passing the issue of polymorphic antigens (Faber et al. Vaccine 2024)
p.12 Lines 471-473 The expansion of animal models relevant for understanding Pv biology and testing interventions will be key for advancing interventions against Pv.
It is unclear what the authors claim here exactly. The characterisation of the P cynomolgi Berok strain in vivo in macaques was carried out in 2019 (Chua et al Nat Comm). Thus the work carried out here has not expanded or established a new animal model. The authors merely adapted existing molecular biology technologies to use in P. cynomolgi.
Further the article should be attentively checked for spelling and grammatical mistakes such as for example:
p 12 line 457-458: “While this transgenic parasite serves as an important step towards in the study of relapsing Plasmodium infections, there are several limitations”
It should be either towards or in
p. 3 Line 103-104 Recently Pc blood stages have been culture adapted should read Recently Pc asexual blood stages have been culture adapted
p 6 Line 249-250 For infection initiated by blood stages parasites , a 200-500 mL parasite stock… should read
For infection initiated by asexual blood stages parasites , a 200-500 µL parasite stock…
p 11 Line 435-436: “A more recent study used CRISPR-based mutation to..” should read Cas-9 mediated mutagenesis….
Comments on the Quality of English LanguageFurther the article should be attentively checked for spelling and grammatical mistakes such as for example:
p 12 line 457-458: “While this transgenic parasite serves as an important step towards in the study of relapsing Plasmodium infections, there are several limitations”
It should be either towards or in
p. 3 Line 103-104 Recently Pc blood stages have been culture adapted should read Recently Pc asexual blood stages have been culture adapted
p 6 Line 249-250 For infection initiated by blood stages parasites , a 200-500 mL parasite stock… should read
For infection initiated by asexual blood stages parasites , a 200-500 µL parasite stock…
p 11 Line 435-436: “A more recent study used CRISPR-based mutation to..” should read Cas-9 mediated mutagenesis….
Reviewer 2 Report
Comments and Suggestions for Authors
The manuscript by Maya A. is well crafted and implemented. They used crispr to generate the swap of Pc CSP for Pv CSP to show no change in measurable viability. IF shows distinct genes.
The PV CSP in PC transmitted to mosquito and monkeys fine.
Great paper with good discussion. This work will open path to testing P. vivax liver vaccines and will be able to answer of CSP plays a role in dormant malaria after invasion. They can test if reduction in hepatocyte infected numbers leads to reduction in relapse from a numbers reduction. This is also a proof of principle to complete lifecycle with swapped genes in P. cynomolgi. P. falciparum csp swap was arrested while Pf CSP works well in the mouse mode.
Reviewer 3 Report
Comments and Suggestions for Authors
Dear authors,
You manuscript “Generation of a transgenic Plasmodium cynomolgi parasite expressing P. vivax circumsporozoite protein for testing pre-erythrocytic malaria vaccines in non-human primates” describes the construction Plasmodium cynomolgi parasite which expresses the P. vivax CSP protein and its characterization.
The manuscript is well written, and contains all necessary parts. However, some issues should be revised before the publication:
My suggestion to shorten the title “Generation of a transgenic Plasmodium cynomolgi parasite expressing P. vivax circumsporozoite protein for testing pre-erythrocytic malaria vaccines in non-human primates” because it is very long, and the use of abbreviations in the species name in the title of the article is unacceptable.
General note - the references in text are given in superscript style - please check if this style is acceptable for this journal; in any case, they are smaller than the main text, which makes them somewhat difficult to read.
L9-10 – “Center for Global Infectious Disease Research, Seattle Children’s Research Institute, Seattle, Washington; CJ current institution: BioNTech SE” – please correct the affiliation.
L17, 19, etc.. – “Pf” please correct to “P. falciparum”, such way of abbreviation is unusual.
L18, 21, etc… – “Pv” – the same.
L25, etc… - “Pc” – the same.
L36 – “Plasmodium vivax; Plasmodium cynomolgi” – should be in italic.
L52 – “P. falciparum (Pf) and P. vivax (Pv)” – such abbreviations are OK, but please correct the issue in Abstract section.
L96 – “reviewed in 27” – please correct such references throughout the text as “reviewed by Author et al [Reference]” – it gives more information and easier to understand.
Section 2.4, 2.5 – It seems that the references to schemes of recombinant plasmid/cassette construction could be useful for readers (they are provided in Results section, but should be mentioned here).
L161 “first amplified and sequenced” and section 2.7 – what kits, equipment and methods have you used? Should be described.
L171 – “three guide sequences” – what are sequences? They should be provided.
L207 – “lines” or “lineages”? What is correct?
L212 – what is it, “SporoCore”? Facility?
L293 – “of a P. cynomolgi” – please correct to “of P. cynomolgi” .
References section – the most references contain no doi, and that could be useful for readers. Please add.
Round 2
Reviewer 1 Report
Comments and Suggestions for Authors
The authors have clarified and detailed their methodology, they have tuned down some of the claims that were part of the original article and chosen a more inclusive contextualization of their work. While the author's argument goes that details of previous work "were omitted from the final version to reduce the scope to a more methods-oriented version of the manuscript", the better contextualization of the new version contributes to the positioning of the paper in the context of molecular tools for vaccine research, which in turn also positively frames the contribution of their paper to the field.
Notably, in response to the comment concerning the contribution of the technology to P vivax vaccine research, the authors argue that they have already used the parasite in in vivo experiments, but that the data will be published in another article: "For example, we have already used this transgenic in a small challenge study to test the protection afforded by a CSP-based Pv vaccine, a challenge study that would not have been possible without this parasite (manuscript in preparation)..."
Since the reviewers did not know this and readers will neither, it would probably be good to refer to this manuscript in preparation to actually support the point that this method contributes to P. vivax vaccine research
Author Response
The authors have clarified and detailed their methodology, they have tuned down some of the claims that were part of the original article and chosen a more inclusive contextualization of their work. While the author's argument goes that details of previous work "were omitted from the final version to reduce the scope to a more methods-oriented version of the manuscript", the better contextualization of the new version contributes to the positioning of the paper in the context of molecular tools for vaccine research, which in turn also positively frames the contribution of their paper to the field.
Notably, in response to the comment concerning the contribution of the technology to P vivax vaccine research, the authors argue that they have already used the parasite in in vivo experiments, but that the data will be published in another article: "For example, we have already used this transgenic in a small challenge study to test the protection afforded by a CSP-based Pv vaccine, a challenge study that would not have been possible without this parasite (manuscript in preparation)..."
Since the reviewers did not know this and readers will neither, it would probably be good to refer to this manuscript in preparation to actually support the point that this method contributes to P. vivax vaccine research.
We sincerely thank the reviewer for their continued efforts to improve our manuscript, and the careful evaluation of our revisions based on the suggestions provided. Based on the introduction, we feel that we have given good context for the need for better preclinical challenge capacities for Pv, and thus that the reader will recognize the value that this transgenic will have in directly testing Pv-CSP vaccines without evidence that it is in use. While we appreciate the suggestion to include in the discussion section a reference to the vaccine efficacy study that we have already performed using this transgenic parasite, we have prefer to not cite manuscripts in preparation, and to instead allow for peer review prior to claiming any contributions to the literature.
Reviewer 3 Report
Comments and Suggestions for Authors
Dear authors,
You did a lot of work, thank you!
All my comments were corrected or reasoned arguments are given.
I have only one remark:
L185 (and elsewhere in the text) - "Pcy c" - can not find in the text what is it. Please, correct.
Author Response
Dear authors,
You did a lot of work, thank you!
All my comments were corrected or reasoned arguments are given.
I have only one remark:
L185 (and elsewhere in the text) - "Pcy c" - can not find in the text what is it. Please, correct.
We sincerely thank the reviewer for their time in carefully reviewing our manuscript and the changes that were incorporated following the first revisions. We have found the typo in L185 and one other location and removed the errant "c"; this is not highlighted in the revised version as it they were removals. Thank you again for the careful attention in reviewing this manuscript.